# Harnessing the Power of Infinitely Wide Deep Nets on Small-data Tasks

**Sanjeev Arora**
Princeton University
arora@cs.princeton.edu

**Simon S. Du**
Institute for Advanced Study
ssdu@ias.edu

**Zhiyuan Li**
Princeton University
zhiyuanli@cs.princeton.edu

**Ruslan Salakhutdinov**
Carnegie Mellon University
rsalakhu@cs.cmu.edu

**Ruosong Wang**
Carnegie Mellon University
ruosongw@andrew.cmu.edu

**Dingli Yu**
Princeton University
dingliy@cs.princeton.edu

## Abstract

Recent research shows that the following two models are equivalent: (a) infinitely wide neural networks (NNs) trained under $\ell_2$ loss by gradient descent with infinitesimally small learning rate (b) kernel regression with respect to so-called Neural Tangent Kernels (NTKs) (Jacot et al., 2018). An efficient algorithm to compute the NTK, as well as its convolutional counterparts, appears in Arora et al. (2019a), which allowed studying performance of infinitely wide nets on datasets like CIFAR-10. However, super-quadratic running time of kernel methods makes them best suited for small-data tasks. We report results suggesting neural tangent kernels perform strongly on low-data tasks.

1. On a standard testbed of classification/regression tasks from the UCI database, NTK SVM beats the previous gold standard, Random Forests (RF), and also the corresponding finite nets.
2. On CIFAR-10 with $10 - 640$ training samples, Convolutional NTK consistently beats ResNet-34 by $1\%$ - $3\%$.
3. On VOC07 testbed for few-shot image classification tasks on ImageNet with transfer learning (Goyal et al., 2019), replacing the linear SVM currently used with a Convolutional NTK SVM consistently improves performance.
4. Comparing the performance of NTK with the finite-width net it was derived from, NTK behavior starts at lower net widths than suggested by theoretical analysis(Arora et al., 2019a). NTK's efficacy may trace to lower variance of output.

## 1 Introduction

Modern neural networks (NNs) have way more parameters than training data points, which allow them to achieve near-zero training error while simultaneously — for some reason yet to be understood — have low generalization error (Zhang et al., 2016). This motivated formal study of highly overparametrized networks, including networks whose width (i.e., number of nodes in layers, or number of channels in convolutional layers) goes to infinity. A recent line of theoretical results shows that with $\ell_2$ loss and infinitesimal learning rate, in the limit of infinite width the trajectory of training converges to kernel regression with a particular kernel, neural tangent kernel (NTK) (Jacot et al., 2018). For convolutional networks, the kernel is CNTK. See Section 2 for more discussions. Arora et al. (2019a) gave an algorithm to exactly compute the kernel corresponding to the infinite limit of various realistic NN architectures with convolutions and pooling layers, allowing them to compute performance on CIFAR-10, which revealed that the infinite networks have 6 to $8\%$ higher error than their finite counterparts. This is still fairly good performance for a fixed kernel.

Ironically, while the above-mentioned analysis, at first sight, appears to reduce the study of a complicated model — deep networks — to an older, simpler model — kernel regression — in practice the simpler model is computationally less efficient because running time of kernel regression can be quadratic in the number of data points![1] Thus computing using CNTK kernel on large datasets like ImageNet currently appears infeasible. Even on CIFAR-10, it seems infeasible to incorporate data augmentation.

However, kernel classifiers are very efficient on small datasets. Here NTKs could conceivably be practical while at the same time bringing some of the power of deep networks to these settings. We recall that recently Olson et al. (2018) showed that multilayer neural networks can be reasonably effective on small datasets, specifically on a UCI testbed of tasks with as few as dozens of training examples. Of course, this required some hyperparameter tuning, although they noted that such tuning is also needed for the champion method, Random Forests (RF), which multilayer neural networks could not beat.

It is thus natural to check if NTK — corresponding to infinitely wide fully-connected networks — performs well in such small-data tasks[2]. Convex objectives arising from kernels have stable solvers with minimal hyperparameter tuning. Furthermore, random initialization in deep network training seems to lead to higher variance in the output, which can hurt performance in small-data settings. Can NTK's do better? Below we will see that in the setup of Olson et al. (2018), NTK predictors indeed outperforms corresponding finite deep networks, and also slightly beats the earlier gold standard, Random Forests. This suggests NTK predictors should belong in any list of off-the-shelf machine learning methods.

Following are low-data settings where we used NTKs and CNTKs:

- In the testbed of 90 classification tasks from UCI database, NTK predictor achieves superior, and arguably the strongest classification performance. This is verified via several standard statistical tests, including Friedman Rank, Average Accuracy, Percentage of the Maximum Accuracy (PMA) and probability of achieving 90%/95% maximum accuracy (P90 and P95), performed to compare performances of different classifiers on 90 datasets from UCI database. (The authors plan to release the code, to allow off-the-shelf use of this method. It does not require GPUs.)
- We find the performance of NN is close to that of NTK. On *every* dataset from UCI database, the difference between the classification accuracy of NN and that of NTK is within $5\%$. On the other hand, on some datasets, the difference between classification accuracy of NN (or NTK) and that of other classifiers like RF can be as high as $20\%$. This indicates in low-data settings, NTK is indeed a good description of NN. Furthermore, we find NTK is more stable (smaller variance), which seems to help it achieve better accuracy on small datasets (cf. Figure 2b).
- CNTK is useful in computer vision tasks with small-data. On CIFAR-10, we compare CNTK with ResNet using 10 - 640 training samples and find CNTK can beat ResNet by $1\% - 3\%$. We further study few-shot image classification task on VOC07 dataset. The standard method is to first use a pre-trained network, e.g., ResNet-50 trained on ImageNet, to extract features and then directly apply a linear classifier on the extracted features (Goyal et al., 2019). Here we replace the linear classifier with CNTK and obtain better classification accuracy in various setups.

**Paper organization.** Section 2 discusses related work. Section 3 reviews the derivation of NTK. Section 4 presents experiments using NN and NTK on UCI datasets. Section 5 presents experiments using CNN and CNTK on small CIFAR-10 datasets. Section 6 presents experiments using CNTK for the few-shot learning setting. Additional technical details are presented in appendix.

## 2 RELATED WORK

Our paper is inspired by Fernández-Delgado et al. (2014) which conducted extensive experiments on UCI dataset. Their conclusion is random forest performs the best, which is followed by the SVM with Gaussian kernel. Therefore, RF may be considered as a reference ("gold-standard") to compare

---

[1] The bottleneck is constructing the kernel, which scales quadratically with the number of data points (Arora et al., 2019a). The regression can take cubic time in the number of data points.

[2] Note that NTKs can also be used in kernel SVMs, which are not known to be equivalent to training infinitely wide networks. Currently, equivalence is only known for ridge regression. We tried both.

with new classifiers. Olson et al. (2018) followed this testing strategy to evaluate the performance of modern neural networks concluding that modern neural networks, even though being highly over-parameterized, still give reasonable performances on these small datasets, though not as strong as RFs. Our paper follows the same testing strategy to evaluate the performance of NTK.

The focus of this paper, neural tangent kernel is induced from a neural network architecture. The connection between infinitely wide neural networks and kernel methods is not new (Neal, 1996; Williams, 1997; Roux & Bengio, 2007; Hazan & Jaakkola, 2015; Lee et al., 2018; Matthews et al., 2018; Novak et al., 2019; Garriga-Alonso et al., 2019; Cho & Saul, 2009; Daniely et al., 2016; Daniely, 2017). However, these kernels correspond to neural network where only the last layer is trained. Neural tangent kernel, first proposed by Jacot et al. (2018), is fundamentally different as NTKs correspond to infinitely wide NNs with all layer being trained. Theoretically, a line of work study the optimization and generalization behavior of ultra-wide NNs (Allen-Zhu et al., 2018b;a; Arora et al., 2019b; Du et al., 2018b;a; Li & Liang, 2018; Zou et al., 2018; Yang, 2019). Recently, Arora et al. (2019a) gave non-asymptotic perturbation bound between the NN predictor trained by gradient descent and the NTK predictor. Empirically, Lee et al. (2019) verified on small scale data, NTK is a good approximation to NN. However, Arora et al. (2019a) showed on large scale dataset, NN can outperform NTK which may due to the effect of finite-width and/or optimization procedure.

Generalization to architectures other than fully-connected NN and CNN are recently proposed (Yang, 2019; Du et al., 2019; Bietti & Mairal, 2019). Du et al. (2019) showed graph neural tangent kernel (GNTK) can achieve better performance than its counter part, graph neural network (GNN), on datasets with up to 5000 samples.

## 3 NEURAL NETWORK AND NEURAL TANGENT KERNEL

Since NTK is induced by a NN architecture, we first define a NN formally. Let $x \in \mathbb{R}^d$ be the input, and denote $g^{(0)}(x) = x$ and $d_0 = d$ for notational convenience. We define an $L$-hidden-layer fully-connected neural network recursively:

$$f^{(h)}(x) = W^{(h)}g^{(h-1)}(x) \in \mathbb{R}^{d_h}, \quad g^{(h)}(x) = \sqrt{\frac{c_\sigma}{d_h}}\sigma\left(f^{(h)}(x)\right) \in \mathbb{R}^{d_h}, \qquad h = 1, 2, \ldots, L,$$
(1)

where $W^{(h)} \in \mathbb{R}^{d_h \times d_{h-1}}$ is the weight matrix in the $h$-th layer ($h \in [L]$), $\sigma : \mathbb{R} \to \mathbb{R}$ is a coordinate-wise activation function, and $c_\sigma$ is a scaling factor.[3] In this paper, for NN we will consider $\sigma$ being ReLU or ELU (Clevert et al., 2015) and for NTK we will only consider kernel functions induced by NNs with ReLU activation. The last layer of the neural network is

$$\begin{aligned} f(w, x) = f^{(L+1)}(x) &= W^{(L+1)} \cdot g^{(L)}(x) \\ &= W^{(L+1)} \cdot \sqrt{\frac{c_\sigma}{d_L}}\sigma\left(W^{(L)} \cdot \sqrt{\frac{c_\sigma}{d_{L-1}}}\sigma\left(W^{(L-1)} \cdots \sqrt{\frac{c_\sigma}{d_1}}\sigma\left(W^{(1)}x\right)\right)\right), \end{aligned}$$
(2)

where $W^{(L+1)} \in \mathbb{R}^{1 \times d_L}$ is the weights in the final layer, and we let $w = \left(W^{(1)}, \ldots, W^{(L+1)}\right)$ be all parameters in the network. All the weights are initialized to be i.i.d. $\mathcal{N}(0, 1)$ random variables. From now on, by NTK initialization we mean a neural network with parameterization defined in Equation 2 with all weighted being initialized to be i.i.d. $\mathcal{N}(0, 1)$.

When the hidden widths $d_1, d_2, \ldots, d_L \to \infty$, certain limiting behavior emerges along the gradient trajectory. Let $x, x' \in \mathbb{R}^d$ be two data points, the covariance kernel of the $h$-th layer's outputs, $\Sigma^{(h)}(x, x') = f^{(h)}(x) \cdot f^{(h)}(x')$, can be recursively defined in an analytical form:

$$\begin{aligned} \Sigma^{(0)}(x, x') &= x^\top x', \\ \Lambda^{(h)}(x, x') &= \begin{pmatrix} \Sigma^{(h-1)}(x, x) & \Sigma^{(h-1)}(x, x') \\ \Sigma^{(h-1)}(x', x) & \Sigma^{(h-1)}(x', x') \end{pmatrix} \in \mathbb{R}^{2 \times 2}, \\ \Sigma^{(h)}(x, x') &= c_\sigma \mathbb{E}_{(u,v)\sim\mathcal{N}\left(0, \Lambda^{(h)}\right)}\left[\sigma\left(u\right)\sigma\left(v\right)\right], \end{aligned}$$
(3)

---

[3]Putting an explicit scaling factor in the definition of NN is typically called NTK parameterization (Jacot et al., 2018; Park et al., 2019). Standard parameterization scheme does not have the explicit scaling factor. The derivation of NTK requires NTK parameterization. In our experiments on NNs, we try both parameterization schemes.

| Classifier | Friedman Rank | Average Accuracy | P90 | P95 | PMA |
|:---:|:---:|:---:|:---:|:---:|:---:|
| NTK | **28.34** | **81.95%±14.10%** | **88.89%** | **72.22%** | **95.72% ±5.17%** |
| NN (He init) | 40.97 | 80.88%±14.96% | 81.11% | 65.56% | 94.34% ±7.22% |
| NN (NTK init) | 38.06 | 81.02%±14.47% | 85.56% | 60.00% | 94.55% ±5.89% |
| RF | 33.51 | 81.56% ±13.90% | 85.56% | 67.78% | 95.25% ±5.30% |
| Gaussian Kernel | 35.76 | 81.03% ± 15.09% | 85.56% | **72.22%** | 94.56% ±8.22% |
| Polynomial Kernel | 38.44 | 78.21% ± 20.30% | 80.00% | 62.22% | 91.29% ±18.05% |

Table 1: Comparisons of different classifiers on 90 UCI datasets. P90/P95: the number of datasets a classifier achieves 90%/95% or more of the maximum accuracy, divided by the total number of datasets. PMA: average percentage of the maximum accuracy.

for $h \in [L]$. Crucially, this analytical form holds not only at the initialization, but also holds during the training (when gradient descent with small learning rate is used as the optimization routine).

Formally, NTK is defined as the limiting gradient kernel

$$\Theta\left(x, x'\right) \triangleq \left\langle \frac{\partial f(w,x)}{\partial w}, \frac{\partial f(w,x')}{\partial w} \right\rangle = \sum_{h=1}^{L+1} \left\langle \frac{\partial f(w,x)}{\partial W^{(h)}}, \frac{\partial f(w,x')}{\partial W^{(h)}} \right\rangle .$$

Again, one can obtain a recursive formula

$$\dot{\Sigma}^{(h)}(x,x') = c_\sigma \mathbb{E}_{(u,v)\sim\mathcal{N}\left(0,\Lambda^{(h)}\right)} \left[\dot{\sigma}(u)\dot{\sigma}(v)\right], h = 1, \ldots, L+1 \tag{4}$$

$$\Theta(x,x') = \sum_{h=1}^{L+1} \left( \Sigma^{(h-1)}(x,x') \cdot \prod_{h'=h}^{L+1} \dot{\Sigma}^{(h')}(x,x') \right), \tag{5}$$

where we let $\dot{\Sigma}^{(L+1)}(x,x') = 1$ for convenience. It is easy to check if we fix the first $L'$ layers and only train the remaining $(L + 1 - L')$ layers, then the resulting NTK is $\Theta(x,x') = \sum_{h=L'+1}^{L+1} \left( \Sigma^{(h-1)}(x,x') \cdot \prod_{h'=h}^{L+1} \dot{\Sigma}^{(h')}(x,x') \right)$. Note when $L' = L$, then the resulting NTK is $\Sigma^{(L)}(x,x')$, which is the NNGP kernel (Lee et al., 2018). $L'$ can be viewed as a hyperparameter of NTK classifier, and in our UCI experiment we tune $L'$. Given a kernel function, one can directly use it for downstream classification tasks (Scholkopf & Smola, 2001).

## 4    EXPERIMENTS ON UCI DATASETS

In this section, we present our experimental results on UCI datasets which follow the setup of Fernández-Delgado et al. (2014) with extensive comparisons of classifiers, including random forest, kernel SVM, multilayer neural networks, etc. Section 4.2 discusses the performance of NTK through detailed comparisons with other classifiers tested by Fernández-Delgado et al. (2014). Section 4.2 compares NTK classifier and the corresponding NN classifier and verifies how similar their predictions are. See Table 6 in Appendix A for a summary of datasets we used. The detailed experiment setup, including the choices the datasets, training / test splitting and ranges of hyperparameters, is described in Appendix A. We note that usual methods of obtaining confidence bounds in these low-data settings are somewhat heuristic.

### 4.1    OVERALL PERFORMANCE COMPARISONS

Table 1 lists the performance of 6 classifiers under various metrics: the three top classifiers identified in Fernández-Delgado et al. (2014), namely, RF, Gaussian kernel and polynomial kernel, along with our new methods NTK, NN with He initialization and NN with NTK initialization. Table 1 shows NTK is the best classifier under all metrics, followed by RF, the best classifier identified in Fernández-Delgado et al. (2014). Now we interpret each metric in more details.

**Friedman Ranking and Average Accuracy.**    NTK is the best (Friedman Rank 28.34, Average Accuracy 81.95%), followed by RF (Friedman Rank 33.51, Average Accuracy 81.56%) and then followed by SVM with Gaussian kernel (Friedman Rank 35.76, Average Accuracy 81.03%). The difference between NTK and RF is significant (-5.17 in Friedman Rank and +0.39% in Average Accuracy), just as the superiority of RF is significant compared to other classifiers as claimed in Fernández-Delgado et al. (2014). NN (with either He initialization or NTK initialization) performs significantly better than the polynomial kernel in terms of the Average Accuracy (80.88% and 81.02% vs. 78.21%), but in terms of Friedman Rank, NN with He initialization is worse than polynomial kernel (40.97 vs. 38.44) and NN with NTK initialization is slightly better than polynomial kernel (38.06 vs. 38.44). On many datasets where most classifiers have similar performances, NN's rank is high as well, whereas on other datasets, NN is significantly better than most classifiers, including SVM with polynomial kernel. Therefore, NN enjoys higher Average Accuracy but suffers higher Friedman Rank. For example, on OZONE dataset, NN with NTK initialization is only 0.25% worse than polynomial kernel but their difference in terms of rank is 56. It is also interesting to see that NN with NTK initialization performs better than NN with He initialization (38.06 vs. 40.97 in Friedman Rank and 81.02% vs. 80.88% in Average Accuracy).

**P90/P95 and PMA.**    This measures, for a given classifier, the fraction of datasets on which it achieves more than 90%/95% of the maximum accuracy among all classifiers. NTK is one of the best classifiers (ties with Gaussian kernel on P95), which shows NTK can consistently achieve superior classification performance across a broad range of datasets. Lastly, we consider the Percentage of the Maximum Accuracy (PMA). NTK achieves the best average PMA followed by RF whose PMA is 0.47% below that of NTK and other classifiers are all below 94.6%. An interesting observation is that NTK, NN with NTK initialization and RF have small standard deviation (5.17%, 5.89% and 5.30%) whereas all other classifiers have much larger standard deviation.

## 4.2    PAIRWISE COMPARISONS

**NTK vs. RF.**    In Figure 1a, we compare NTK with RF. There are 42 datasets that NTK outperforms RF and 40 datasets that RF outperforms NTK.[4] The mean difference is 2.51%, which is statistically significant by a Wilcoxon signed rank test. We see NTK and RF perform similarly when the Bayes error rate is low with NTK being slightly better. There are some exceptions. For example, on the BALANCE-SCALE dataset, NTK achieves 98% accuracy whereas RF only achieves 84.1%. When the Bayes error rate is high, either classifier can be significantly better than the other.

**NTK vs. Gaussian Kernel.**    The gap between NTK and Gaussian kernel is more significant. As shown in Figure 1b, NTK generally performs better than Gaussian kernel no matter Bayes error rate is low or high. There are 43 datasets that NTK outperforms Gaussian kernel and there are 34 datasets that Gaussian kernel outperforms NTK. The mean difference is 2.22%, which is also statistically significant by a Wilcoxon signed rank test. On BALLOONS, HEART-SWITZERLAND, PITTSBURG-BRIDGES-MATERIAL, TEACHING, TRAINS datasets, NTK is better than Gaussian kernel by at least 11% in terms of accuracy. These five datasets all have less than 200 samples, which shows that NTK can perform much better than Gaussian kernel when the number of samples is small.

**NTK vs. NN**    In this section we compare NTK with NN. The goals are (i) comparing the performance and (ii) verifying NTK is a good approximation to NN.

In Figure 2a, we compare the performance between NTK and NN with He initialization. For most datasets, these two classifiers perform similarly. However, there are a few datasets that NTK performs significantly better. There are 50 datasets that NTK outperforms NN with He initialization and there are 27 datasets that NN with He initialization outperforms NTK. The mean difference is 1.96%, which is also statistically significant by a Wilcoxon signed rank test.

In Figure 2b, we compare the performance between NTK and NN with NTK initialization. Recall that for NN with NTK initialization, when the width goes to infinity, the predictor is just NTK.

---

[3]The average rank of a classifier over 90 datasets.

[4]We consider classifier A outperform classifier B if the classification accuracy of A is at least 0.1% better than B. The number 0.1% is chosen because results in Fernández-Delgado et al. (2014) only kept the precision up to 0.1%.

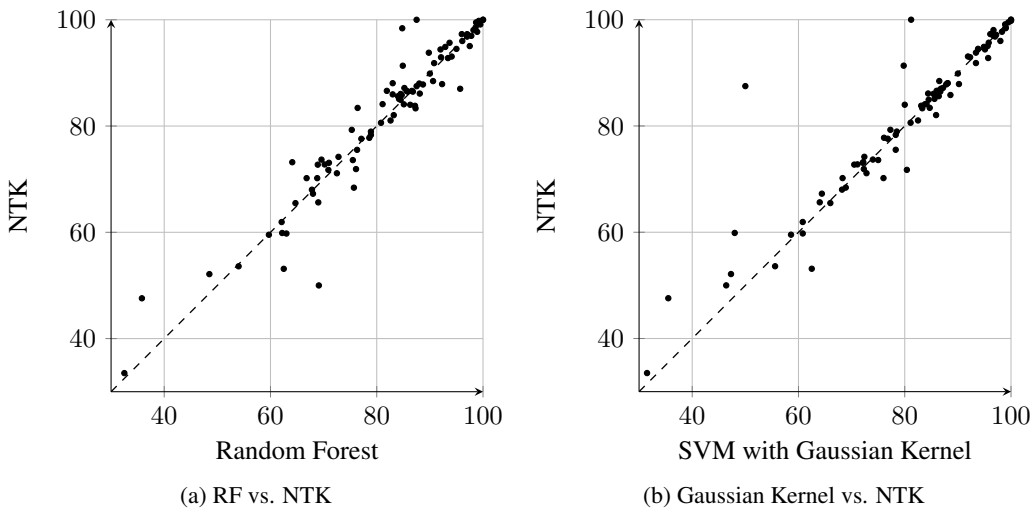

(a) RF vs. NTK

(b) Gaussian Kernel vs. NTK

Figure 1: Performance comparisons between NTK and other classifiers.

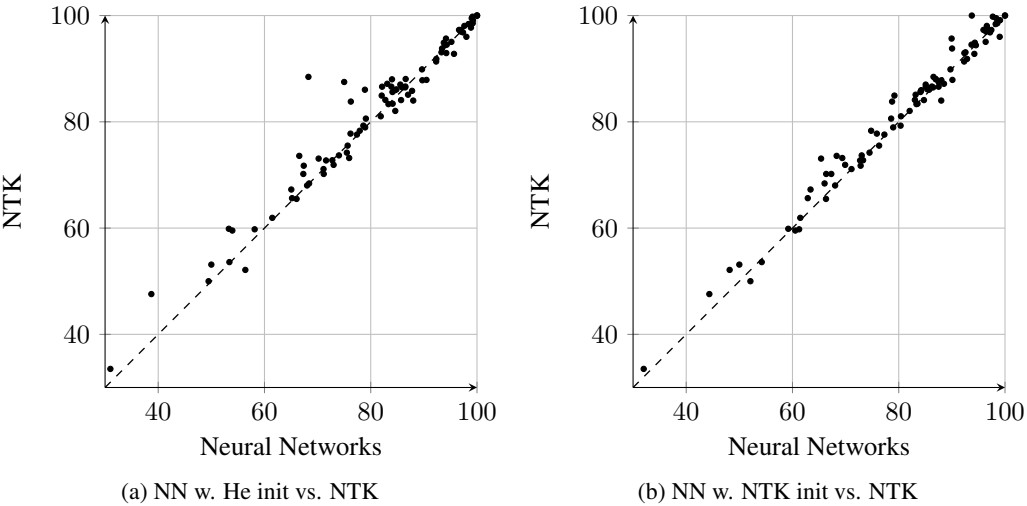

(a) NN w. He init vs. NTK

(b) NN w. NTK init vs. NTK

Figure 2: Performance Comparisons between NTK and NN.

Therefore, we expect these two predictors give similar performance. Figure 2b verifies our conjecture. There is *no* dataset the one classifier is significantly better than the other. We do not have the same observation in Figure 1a, 1b or 2a. Nevertheless, NTK often performs better than NN. There are 52 datasets that NTK outperforms NN with He initialization and there are 25 datasets NN with He initialization outperforms NTK. The mean difference is 1.54%, which is also statistically significant by a Wilcoxon signed rank test.

## 5   EXPERIMENTS ON SMALL CIFAR-10 DATASET

In this section, we study the performance of CNTK on subsampled CIFAR-10 dataset. We randomly choose $n$ samples from CIFAR-10 training set, use them to train CNTK and ResNet-34, and test both classifiers on the whole test set. In our experiments, we vary $n$ from 10 to 1280, and the number of convolutional layers of CNTK varies from 5 to 14. See Appendix B for detailed experiment setup.

The results are reported in Table 2. It can be observed that in this setting CNTK consistently outperforms ResNet-34. The largest gap occurs at $n = 320$ where 14-layer CNTK achieves 36.57% accuracy and ResNet achieves 33.15%. The smallest improvement occurs at $n = 10$: 15.33% vs.

| $n$ | ResNet | 5-layer CNTK | 8-layer CNTK | 11-layer CNTK | 14-layer CNTK |
|---|---|---|---|---|---|
| 10 | 14.59% ± 1.99% | 15.08% ± 2.43% | 15.24% ± 2.44% | 15.31% ± 2.38% | **15.33% ± 2.43%** |
| 20 | 17.50% ± 2.47% | 18.03% ± 1.91% | 18.50% ± 2.03% | 18.69% ± 2.07% | **18.79% ± 2.13%** |
| 40 | 19.52% ± 1.39% | 20.83% ± 1.68% | 21.07% ± 1.80% | 21.23% ± 1.86% | **21.34% ± 1.91%** |
| 80 | 23.32% ± 1.61% | 24.82% ± 1.75% | 25.18% ± 1.80% | 25.40% ± 1.84% | **25.48% ± 1.91%** |
| 160 | 28.30% ± 1.38% | 29.63% ± 1.13% | 30.17% ± 1.11% | 30.46% ± 1.15% | **30.48% ± 1.17%** |
| 320 | 33.15% ± 1.20% | 35.26% ± 0.97% | 36.05% ± 0.92% | 36.44% ± 0.91% | **36.57% ± 0.88%** |
| 640 | 41.66% ± 1.09% | 41.24% ± 0.78% | 42.10% ± 0.74% | 42.44% ± 0.72% | **42.63% ± 0.68%** |
| 1280 | **49.14% ± 1.31%** | 47.21% ± 0.49% | 48.22% ± 0.49% | 48.67% ± 0.57% | 48.86% ± 0.68% |

Table 2: Performance of ResNet-34 and CNTK on small CIFAR-10 Dataset.

| $k$ | linear SVM | 1-layer CNTK | 2-layer CNTK | 3-layer CNTK | 4-layer CNTK | 5-layer CNTK | 6-layer CNTK |
|---|---|---|---|---|---|---|---|
| 1 | 49.46 ± 4.71 | **50.04 ± 4.28** | 49.94 ± 4.19 | 49.66 ± 4.19 | 49.24 ± 4.20 | 48.66 ± 4.26 | 48.00 ± 4.41 |
| 2 | 64.39 ± 5.02 | 65.81 ± 5.28 | **65.89 ± 5.23** | 65.65 ± 5.17 | 65.30 ± 5.10 | 64.76 ± 5.07 | 64.12 ± 5.00 |
| 3 | 70.38 ± 3.28 | **71.60 ± 3.51** | 71.56 ± 3.65 | 71.32 ± 3.74 | 70.96 ± 3.84 | 70.51 ± 3.92 | 69.98 ± 3.95 |
| 4 | 72.43 ± 3.21 | **73.65 ± 3.29** | 73.59 ± 3.28 | 73.35 ± 3.28 | 73.00 ± 3.20 | 72.56 ± 3.20 | 72.09 ± 3.20 |
| 5 | 74.49 ± 3.08 | **75.87 ± 2.88** | 75.69 ± 2.89 | 75.45 ± 2.89 | 75.13 ± 2.85 | 74.68 ± 2.88 | 74.24 ± 2.85 |
| 6 | 76.59 ± 2.15 | **77.64 ± 2.37** | 77.57 ± 2.43 | 77.39 ± 2.50 | 77.13 ± 2.56 | 76.87 ± 2.60 | 76.52 ± 2.63 |
| 7 | 77.11 ± 1.25 | **78.64 ± 1.35** | 78.57 ± 1.39 | 78.39 ± 1.41 | 78.12 ± 1.42 | 77.80 ± 1.41 | 77.43 ± 1.41 |
| 8 | 79.57 ± 0.87 | **80.42 ± 0.88** | 80.38 ± 0.86 | 80.23 ± 0.86 | 79.99 ± 0.88 | 79.72 ± 0.89 | 79.41 ± 0.92 |

Table 3: Performance of linear SVM and CNTK with different number of convolutional layers on the 11-20th classes in VOC07. The cost value $C$ is tuned on the first 10 classes. Feature extracted from `conv5` in ResNet-50.

14.59%. When the number of training data is large, i.e., $n = 1280$, ResNet can outperform CNTK. It is also interesting to see that 14-layer CNTK is the best performing CNTK for all values of $n$, suggesting depth has a significant effect on this task.

## 6 EXPERIMENTS ON FEW-SHOT LEARNING

In this section, we test the ability of NTK as a drop-in classifier to replace the linear classifier in the few-shot learning setting. Using linear SVM on top of extracted features is arguably the most widely used strategy in few-shot learning as linear SVM is easy and fast to train whereas more complicated strategies like fine-tuning or training a neural network on top of the extracted feature have more randomness and may overfit due to the small size of the training set. NTK has the same benefits (easy and fast to train, no randomness) as the linear classifier but also allows some non-linearity in the design. Note since we consider image classification tasks, we use CNTK in experiments in this section.

**Experiment Setup.** We mostly follow the settings of Goyal et al. (2019). Features are extracted from layer `conv1, conv2, conv3, conv4, conv5` in ResNet-50 (He et al., 2016) trained on ImageNet (Deng et al., 2009), and we use these features for VOC07 classification task. The number of positive examples $k$ varies from 1 to 8. For each value of $k$, for each class in the VOC07 dataset, we choose $k$ positive examples and $19k$ negative examples. For each $k \in \{1, 2, \ldots, 8\}$ and for each class, we randomly choose 10 independent sets with $20k$ training samples in each set. We

| $k$ | linear SVM | 1-layer CNTK | 2-layer CNTK | 3-layer CNTK | 4-layer CNTK | 5-layer CNTK | 6-layer CNTK |
|---|---|---|---|---|---|---|---|
| 1 | 23.25 ± 2.89 | 24.54 ± 3.39 | **24.57 ± 3.46** | **24.57 ± 3.50** | 24.52 ± 3.51 | 24.43 ± 3.50 | 24.34 ± 3.49 |
| 2 | 32.26 ± 3.85 | **33.81 ± 4.25** | 33.77 ± 4.33 | 33.68 ± 4.40 | 33.53 ± 4.46 | 33.31 ± 4.51 | 33.04 ± 4.55 |
| 3 | 38.41 ± 2.74 | 40.06 ± 2.70 | **40.10 ± 2.47** | 40.02 ± 2.41 | 39.87 ± 2.37 | 39.65 ± 2.35 | 39.36 ± 2.29 |
| 4 | 39.88 ± 2.08 | **42.20 ± 1.78** | 41.80 ± 2.13 | 42.64 ± 2.00 | 42.51 ± 2.02 | 42.32 ± 2.03 | 42.08 ± 2.03 |
| 5 | 42.46 ± 2.70 | **44.71 ± 3.12** | 44.69 ± 2.98 | 44.72 ± 3.01 | 44.69 ± 3.04 | 44.57 ± 3.07 | 44.40 ± 3.10 |
| 6 | 45.71 ± 3.27 | **49.02 ± 2.63** | 48.63 ± 2.69 | 48.63 ± 2.68 | 48.92 ± 2.71 | 48.79 ± 2.74 | 48.63 ± 2.77 |
| 7 | 47.97 ± 3.25 | **50.89 ± 3.50** | 50.43 ± 3.48 | 50.50 ± 3.47 | 50.48 ± 3.44 | 50.39 ± 3.44 | 50.24 ± 3.41 |
| 8 | 49.81 ± 2.18 | **52.32 ± 2.65** | 52.06 ± 2.32 | 52.23 ± 2.38 | 52.31 ± 2.40 | 52.26 ± 2.45 | 52.14 ± 2.50 |

Table 4: Performance of linear SVM and CNTK with different number of convolutional layers on the 11-20th classes in VOC07. The cost value $C$ is tuned on the first 10 classes. Feature extracted from `conv4` in ResNet-50.

| $k$ | linear SVM | 1-layer CNTK | 2-layer CNTK | 3-layer CNTK | 4-layer CNTK | 5-layer CNTK | 6-layer CNTK |
|---|---|---|---|---|---|---|---|
| 1 | $14.38 \pm 1.98$ | $\mathbf{15.42 \pm 1.90}$ | $15.36 \pm 1.93$ | $15.30 \pm 1.97$ | $15.24 \pm 2.02$ | $15.17 \pm 2.05$ | $15.10 \pm 2.08$ |
| 2 | $16.67 \pm 1.50$ | $\mathbf{18.48 \pm 1.58}$ | $18.39 \pm 1.55$ | $18.28 \pm 1.52$ | $18.18 \pm 1.49$ | $18.09 \pm 1.47$ | $17.99 \pm 1.45$ |
| 3 | $19.56 \pm 1.04$ | $\mathbf{21.79 \pm 1.69}$ | $21.73 \pm 1.63$ | $21.67 \pm 1.60$ | $21.61 \pm 1.55$ | $21.54 \pm 1.50$ | $21.47 \pm 1.46$ |
| 4 | $20.53 \pm 1.62$ | $\mathbf{23.39 \pm 2.13}$ | $23.36 \pm 2.18$ | $23.29 \pm 2.22$ | $23.20 \pm 2.24$ | $23.11 \pm 2.26$ | $23.03 \pm 2.27$ |
| 5 | $21.51 \pm 1.68$ | $\mathbf{25.13 \pm 2.36}$ | $25.09 \pm 2.37$ | $25.03 \pm 2.36$ | $24.96 \pm 2.36$ | $24.87 \pm 2.36$ | $24.78 \pm 2.37$ |
| 6 | $23.51 \pm 2.39$ | $\mathbf{26.51 \pm 2.39}$ | $26.26 \pm 2.43$ | $26.10 \pm 2.47$ | $25.97 \pm 2.47$ | $25.87 \pm 2.47$ | $25.78 \pm 2.46$ |
| 7 | $24.33 \pm 1.59$ | $27.98 \pm 2.46$ | $27.75 \pm 2.52$ | $\mathbf{28.24 \pm 2.22}$ | $28.20 \pm 2.22$ | $28.14 \pm 2.21$ | $28.08 \pm 2.21$ |
| 8 | $25.31 \pm 2.07$ | $27.76 \pm 2.87$ | $\mathbf{28.52 \pm 2.65}$ | $\mathbf{28.52 \pm 2.69}$ | $28.49 \pm 2.70$ | $28.44 \pm 2.72$ | $28.38 \pm 2.74$ |

Table 5: Performance of linear SVM and CNTK with different number of convolutional layers on the 11-20th classes in VOC07. The cost value $C$ is tuned on the first 10 classes. Feature extracted from `conv3.` in ResNet-50

report the mean and standard deviation of mAP on the test split of VOC07 dataset. This setting has been used in numbers of previous few-shot learning papers (Goyal et al., 2019; Zhang et al., 2017).

We take the extracted features as the input to CNTK. We tried CNTK with 0-6 convolution layers, a global average pooling layer and a fully-connected layer. We normalize the data in the feature space, and finally use SVM to train the classifiers. Note without the convolution layer, it is equivalent to directly applying linear SVM after global average pooling and normalization. We use `sklearn.svm.LinearSVC` to train linear SVMs, and `sklearn.svm.SVC` to train kernel SVMs (for CNTK). To train SVM, we choose the cost value $C$ from $2^{[-19,-4]} \cup 10^{[-7,2]}$ and set the `class_weight` ratio to be $2 : 1$ for positive/negative classes as in Goyal et al. (2019).

Since the number of given samples is usually small, Goyal et al. (2019) chooses to report the performance of the best cost value $C$. In our experiments, we use a more standard method to perform cross-validation. We use the first 10 classes of VOC07 to tune $C$, and report the performance of selected $C$ in the other 10 classes in Table 3-5. We also report the performance of the best $C$ as in Goyal et al. (2019), in Tables 7-9 in the appendix for completeness.

**Discussions.** First, we find CNTK is a strong drop-in classifier to replace the linear classifier in few-shot learning. Tables 3-9 clearly demonstrate that CNTK is consistently better than linear classifier. Note Tables 3-9 only show the prediction accuracy in an average sense. In fact, we find that on every randomly sampled training set, CNTK always gives a better performance. We conjecture that CNTK gives better performance because the non-linearity in CNTK helps prediction. This is verified by looking at the performance gain of CNTK for different feature extractors. Note `Conv5` is often considered to be most useful for linear classification. There, CNTK only outperforms the linear classifier by about $1\%$. On the other hand, with `Conv3` and `Conv4` features, CNTK can outperform linear classifier by around $2\%$ and sometimes by $3\%$–$4\%$ (last two rows in Table 5). We believe this happens because `Conv3` and `Conv4` correspond to middle-level features, and thus non-linearity is indeed beneficial for better accuracy.

We also observe that CNTK often performs the best with a single convolutional layer. This is expected since features extracted by ResNet-50 already produce a good representation. Nevertheless, we find for middle-level features `Conv3` with $k = 7$ or 8, CNTK with 3 convolutional layers give the best performance. We believe this is because with more data, utilizing the non-linearity induced by CNTK can further boost the performance.

## 7 CONCLUSION

The Neural Tangent Kernel, discovered by mathematical curiosity about deep networks in the limit of infinite width, is found to yield superb performance on low-data tasks, beating extensively tuned versions of classic methods such as random forests. The (fully-connected) NTK classifiers are easy to compute (no GPU required) and thus should be a good off-the-shelf classifier in many settings. We plan to release our code to allow drop-in replacement for SVMs and linear regression.

Many theoretical questions arise. Do NTK SVMs correspond to some infinite net architecture (as NTK ridge regression does)? What explains generalization in small-data settings? (This understanding is imperfect even for random forests.) Finally one can derive NTK corresponding to other

architectures, e.g., recurrent neural tangent kernel (RNTK) induced by recurrent neural networks. It would be an interesting future research direction to test their performance on benchmark tasks.

## ACKNOWLEDGMENTS

S. Arora, Z. Li and D. Yu are supported by NSF, ONR, Simons Foundation, Schmidt Foundation, Amazon Research, DARPA and SRC. S. S. Du is supported by National Science Foundation (Grant No. DMS-1638352) and the Infosys Membership. R. Salakhutdinov and R. Wang are supported in part by NSF IIS-1763562, AFRL CogDeCON FA875018C0014, and DARPA SAGAMORE HR00111990016. Part of this work was done while S. S. Du was visiting Google Brain Princeton and R. Wang was visiting Princeton University. The authors would like to thank Amazon Web Services for providing compute time for the experiments in this paper, and NVIDIA for GPU support. We thank Priya Goyal for providing experiment details of Goyal et al. (2019). We thank Xiaolong Wang for discussing the few-shot learning task.

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

## A    ADDITIONAL EXPERIMENTAL DETAILS ON UCI

In this section, we describe our experiment setup.

**Dataset Selection**    Since we only wish to test on small datasets, we only select UCI datasets with number of samples smaller than 5000. Furthermore, we only use datasets without explicit training / testing splitting from the link: `http://persoal.citius.usc.es/manuel.fernandez.delgado/papers/jmlr/data.tar.gz`, which are the pre-processed datasets by Fernández-Delgado et al. (2014). These datasets are originally from UCI datasets (including most of the datasets before March, 2013) and 4 real-world datasets not included in the UCI repository (see Fernández-Delgado et al. (2014) for details). The datasets are pre-processed to be classification problems and all categorical features are turned into numerical features and normalized along the samples for each feature. The reason to discard datasets with explicit splitting is that we found there is an obvious distributional shift between training and testing data. The pre-processed data provided from the link shows that training and testing data are normalized with different mean and standard deviation. For example, on the AUDIOLOGY-STD dataset, the support of the first feature is $\{-0.761526, 1.30547\}$ in training and $\{-1.2, 0.8\}$ in testing; on the ANNEALING dataset, the support of the first feature is $\{-0.670274, -0.223425, 3.79822\}$ in training and $\{-0.111111, 0, 1\}$ in testing. See Table 6 for a summary of the datasets.

|       | # samples | # features | # classes |
|-------|-----------|------------|-----------|
| min   | 10        | 3          | 2         |
| 25%   | 178       | 8          | 2         |
| 50%   | 583       | 16         | 3         |
| 75%   | 1022      | 32         | 6         |
| max   | 5000      | 262        | 100       |

Table 6: Dataset Summary

**Performance Comparison Details**    We follow the comparison setup in Fernández-Delgado et al. (2014) that we report 4-fold cross-validation. For hyperparameters, we tune them with the same validation methodology in Fernández-Delgado et al. (2014): all available training samples are randomly split into one training and one test set, while imposing that each class has the same number of training and test samples. Then the parameter with best validation accuracy is selected. It is possible to give confidence bounds for this parameter tuning scheme, but they are worse than standard ones for separated training/validation/testing data. For NTK and NN classifiers we train them on these 90 datasets. For other classifiers, we use the results from Fernández-Delgado et al. (2014).

**NTK Specification**    We calculate NTK induced fully-connected neural networks with $L$ layers where $L'$ bottom layers are fixed, and then use $C$-support vector classification implemented by `sklearn.svm`. We tune hyperparameters $L$ from 1 to 5, $L'$ from 0 to $L-1$, and cost value $C$ as powers of ten from $-2$ to 4. The number of kernels used is 15, so the total number of parameter combinations is 105. Note this number is much less than the number of hyperparameter of Gaussian Kernel reported in Fernández-Delgado et al. (2014) where they tune hyperparameters with 500 combinations.

**NN Specification**    We use fully-connected NN with $L$ layers, 512 number of hidden nodes per layer and use gradient descent to train the neural network. We tune hyperparameters $L$ from 1 to 5, with / without batch normalization and learning rate 0.1 or 1. We run gradient descent for 2000 epochs.[5] We treat NN with He initialization and NTK initialization as two classifiers and report their results separately.

---

[5]We found NN can achieve 0 training loss after less than 100 epochs. This is consistent with the observation in Olson et al. (2018).

## B    ADDITIONAL EXPERIMENTAL DETAILS ON SMALL CIFAR-10 DATASETS

We randomly choose $n/10$ samples from each class of CIFAR-10 training set and test classifiers on the whole testing set. $n$ varies from 10 to 1280. For each $n$, we repeat 20 times and report the mean accuracy and its standard deviation for each classifier.

The number of convolution layers of CNTK ranges from 5-14. After convolutional layers, we apply a global pooling layer and a fully connected layer. We refer readers to Arora et al. (2019a) for exact formulas of CNTK. We normalize the kernel such that each sample has unit length in feature space.

We use ResNet-34 with width 64,128,256 and default hyperparameters: learning rate 0.1, momentum 0.9, weight decay 0.0005. We decay the learning rate by 10 at the epoch of 80 and 120, with 160 training epochs in total. The training batch size is the minimum of the size of the whole training dataset and 160. We report the best testing accuracy among epochs.

## C    ADDITIONAL RESULTS IN FEW-SHOT LEARNING

Tables 7-9 show the performance of the best $C$ as has been done in Goyal et al. (2019).

| $k$ | linear SVM | 1-layer CNTK | 2-layer CNTK | 3-layer CNTK | 4-layer CNTK | 5-layer CNTK | 6-layer CNTK |
|---|---|---|---|---|---|---|---|
| 1 | $52.63 \pm 5.35$ | $\mathbf{53.16 \pm 4.96}$ | $53.08 \pm 4.89$ | $52.83 \pm 4.85$ | $52.44 \pm 4.85$ | $51.88 \pm 4.85$ | $51.21 \pm 4.88$ |
| 2 | $65.08 \pm 2.69$ | $66.25 \pm 3.09$ | $\mathbf{66.29 \pm 3.09}$ | $66.11 \pm 3.07$ | $65.79 \pm 3.04$ | $65.33 \pm 3.00$ | $64.77 \pm 2.99$ |
| 3 | $71.78 \pm 1.85$ | $\mathbf{72.76 \pm 1.76}$ | $72.72 \pm 1.73$ | $72.49 \pm 1.71$ | $72.14 \pm 1.73$ | $71.71 \pm 1.76$ | $71.17 \pm 1.78$ |
| 4 | $74.40 \pm 1.86$ | $\mathbf{75.37 \pm 1.95}$ | $75.33 \pm 1.95$ | $75.14 \pm 1.96$ | $74.85 \pm 1.96$ | $74.48 \pm 1.99$ | $74.05 \pm 1.97$ |
| 5 | $75.94 \pm 2.51$ | $\mathbf{77.01 \pm 2.39}$ | $76.91 \pm 2.39$ | $76.72 \pm 2.39$ | $76.47 \pm 2.37$ | $76.13 \pm 2.39$ | $75.76 \pm 2.39$ |
| 6 | $76.39 \pm 1.27$ | $\mathbf{77.15 \pm 1.47}$ | $77.14 \pm 1.50$ | $77.02 \pm 1.55$ | $76.83 \pm 1.58$ | $76.62 \pm 1.60$ | $76.35 \pm 1.61$ |
| 7 | $78.18 \pm 0.86$ | $\mathbf{79.42 \pm 0.85}$ | $79.36 \pm 0.87$ | $79.22 \pm 0.88$ | $79.00 \pm 0.91$ | $78.75 \pm 0.93$ | $78.50 \pm 0.94$ |
| 8 | $79.78 \pm 0.65$ | $\mathbf{80.47 \pm 0.59}$ | $80.45 \pm 0.61$ | $80.33 \pm 0.63$ | $80.14 \pm 0.65$ | $79.92 \pm 0.67$ | $79.66 \pm 0.70$ |

Table 7: Performance of linear SVM and CNTK with different number of convolutional layers on all classes in VOC07 with the best $C$. Feature extracted from `conv5` in ResNet-50

| $k$ | linear SVM | 1-layer CNTK | 2-layer CNTK | 3-layer CNTK | 4-layer CNTK | 5-layer CNTK | 6-layer CNTK |
|---|---|---|---|---|---|---|---|
| 1 | $24.18 \pm 2.64$ | $\mathbf{25.15 \pm 2.93}$ | $25.13 \pm 3.00$ | $25.06 \pm 3.05$ | $24.95 \pm 3.10$ | $24.80 \pm 3.13$ | $24.60 \pm 3.14$ |
| 2 | $31.59 \pm 2.64$ | $\mathbf{32.88 \pm 2.91}$ | $32.81 \pm 2.94$ | $32.67 \pm 2.96$ | $32.48 \pm 2.98$ | $32.24 \pm 2.99$ | $31.94 \pm 3.01$ |
| 3 | $37.34 \pm 2.18$ | $\mathbf{38.69 \pm 2.39}$ | $38.64 \pm 2.26$ | $38.53 \pm 2.28$ | $38.36 \pm 2.31$ | $38.12 \pm 2.33$ | $37.83 \pm 2.35$ |
| 4 | $39.67 \pm 2.31$ | $41.34 \pm 2.33$ | $\mathbf{41.39 \pm 2.28}$ | $41.33 \pm 2.24$ | $41.21 \pm 2.22$ | $41.02 \pm 2.20$ | $40.77 \pm 2.18$ |
| 5 | $41.74 \pm 1.20$ | $\mathbf{43.48 \pm 1.40}$ | $43.48 \pm 1.24$ | $43.44 \pm 1.25$ | $43.32 \pm 1.27$ | $43.12 \pm 1.30$ | $42.89 \pm 1.33$ |
| 6 | $44.68 \pm 1.84$ | $\mathbf{46.77 \pm 1.82}$ | $46.42 \pm 1.91$ | $46.44 \pm 1.90$ | $46.39 \pm 1.91$ | $46.25 \pm 1.93$ | $46.08 \pm 1.94$ |
| 7 | $47.71 \pm 1.84$ | $\mathbf{50.07 \pm 1.97}$ | $49.74 \pm 1.95$ | $49.63 \pm 1.81$ | $49.56 \pm 1.81$ | $49.41 \pm 1.82$ | $49.19 \pm 1.81$ |
| 8 | $48.91 \pm 1.90$ | $\mathbf{51.32 \pm 1.97}$ | $50.98 \pm 1.87$ | $51.09 \pm 1.84$ | $51.09 \pm 1.81$ | $51.00 \pm 1.78$ | $50.84 \pm 1.76$ |

Table 8: Performance of linear SVM and CNTK with different number of convolutional layers on all classes in VOC07 with the best $C$. Feature extracted from `conv4` in ResNet-50

| $k$ | linear SVM | 1-layer CNTK | 2-layer CNTK | 3-layer CNTK | 4-layer CNTK | 5-layer CNTK | 6-layer CNTK |
|---|---|---|---|---|---|---|---|
| 1 | $14.72 \pm 1.49$ | $\mathbf{15.01 \pm 1.27}$ | $14.96 \pm 1.28$ | $14.92 \pm 1.30$ | $14.87 \pm 1.32$ | $14.81 \pm 1.34$ | $14.75 \pm 1.36$ |
| 2 | $16.67 \pm 1.04$ | $\mathbf{17.52 \pm 1.05}$ | $17.45 \pm 1.03$ | $17.36 \pm 1.02$ | $17.28 \pm 1.00$ | $17.20 \pm 0.99$ | $17.11 \pm 0.97$ |
| 3 | $19.21 \pm 1.31$ | $\mathbf{20.65 \pm 1.68}$ | $20.63 \pm 1.65$ | $20.58 \pm 1.63$ | $20.53 \pm 1.61$ | $20.47 \pm 1.59$ | $20.41 \pm 1.57$ |
| 4 | $20.87 \pm 1.49$ | $\mathbf{22.28 \pm 1.80}$ | $22.27 \pm 1.81$ | $22.22 \pm 1.83$ | $22.16 \pm 1.83$ | $22.09 \pm 1.83$ | $22.03 \pm 1.82$ |
| 5 | $21.98 \pm 1.17$ | $\mathbf{23.45 \pm 1.14}$ | $23.45 \pm 1.12$ | $23.43 \pm 1.08$ | $23.38 \pm 1.07$ | $23.32 \pm 1.05$ | $23.25 \pm 1.04$ |
| 6 | $23.02 \pm 1.56$ | $\mathbf{24.44 \pm 1.61}$ | $24.29 \pm 1.66$ | $24.27 \pm 1.82$ | $24.25 \pm 1.81$ | $24.21 \pm 1.80$ | $24.18 \pm 1.78$ |
| 7 | $24.38 \pm 1.52$ | $25.96 \pm 1.80$ | $25.97 \pm 1.70$ | $\mathbf{25.98 \pm 1.70}$ | $25.96 \pm 1.69$ | $25.92 \pm 1.67$ | $25.88 \pm 1.66$ |
| 8 | $25.28 \pm 1.39$ | $27.06 \pm 1.61$ | $27.15 \pm 1.58$ | $\mathbf{27.18 \pm 1.56}$ | $27.17 \pm 1.53$ | $27.13 \pm 1.51$ | $27.09 \pm 1.50$ |

Table 9: Performance of linear SVM and CNTK with different number of convolutional layers on all classes in VOC07 with the best $C$. Feature extracted from `conv3` in ResNet-50

