# OpenReview forum: "Harnessing the Power of Infinitely Wide Deep Nets on Small-data Tasks"
_ICLR.cc/2020/Conference — Accept (Spotlight)_

### Official Review · AnonReviewer1 · 2019-10-20
**Official Blind Review #1**

**Rating:** 8

**Review:**

[Summary]
This paper performs an extensive empirical evaluation of Neural Tangent Kernel (NTK) classifiers---kernel methods that theoretically characterize infinitely wide neural nets---on small-data tasks. Experiments show that NTK classifiers (1) strongly resemble the performance of neural nets on small-data tasks, (2) can beat prior benchmark methods such as Random Forests (RF) on classification tasks in the UCI dataset, and (3) can also outperform standard linear SVM on a few-shot learning task.

[Pros]
The question considered in this paper is well motivated, and a very natural extension of Lee et al. (2019) and Arora et al. (2019a). These papers show that NTK performs well on (relatively) large benchmark tasks such as CIFAR-10 but is still a bit inferior to fully trained neural nets. On the other hand, for small-data tasks, the relationship is reversed --- neural nets are slightly inferior to more traditional methods such as random forests (e.g. from Fernandez-Delgado et al. 2014) and Gaussian kernel SVMs. As the NTK gives a limiting characterization for wide neural nets, it is a sensible question to test the performance of NTK on these small datasets, and see if they can improve over neural nets and compare more favorably against the traditional methods.

The experimental results, in my perspective, is a reasonably convincing evidence that the resemblance between NTK and NN on small-data tasks is stronger than on larger tasks such as CIFAR-10, which agrees with the NTK theory. In addition to the UCI datasets, the paper also tries out NTK in a few-shot learning task and show that SVM with the convolutional NTK does better than the linear SVM as the few-shot learner. I am less familiar with few-shot learning though so am not entirely sure about the strength of this part.

The paper is well-written and delivers its messages clearly. The results and discussions are easy to follow.

[Cons, and suggestions]
The message that “NTK beats RF” seems a bit delicate to me, specifically considering the fact that the average accuracies of (NTK, NN, RF) are all pretty close but the Friedman rank comparison says NTK > RF > NN (somewhat more significantly). This implies the difference between all these methods has to be small and it’s only that NTK happens to win on more tasks. In addition, NTK tunes one more parameter (L’) than NNs, so I guess perhaps NNs can also be tuned to outperform RF in the rank sense if we also tune L’ (by fixing the bottom L’ layers to be not trained) in NNs?

Also, it would be better if the authors could provide a bit more background on the metrics used in the UCI experiments -- for example, the Friedman rank is not defined in the paper.

**Experience Assessment:**

I have published one or two papers in this area.

**Review Assessment: Checking Correctness Of Derivations And Theory:**

N/A

**Review Assessment: Checking Correctness Of Experiments:**

I carefully checked the experiments.

**Review Assessment: Thoroughness In Paper Reading:**

I read the paper at least twice and used my best judgement in assessing the paper.

---

> ### Author Response · Authors · 2019-11-15
> **Response**
>
> Thank you for your positive review. We have revised our paper according to your suggestion. Regarding your comment “NTK tunes one more parameter (L’) than NNs…”: Since training all layers in NN is the standard practice, we did not fix the first $L’$ layers. Also note that for experiments on UCI, more hyper-parameters do not necessarily give better performance because we used 4-fold cross-validation.

---

### Official Review · AnonReviewer3 · 2019-10-23
**Official Blind Review #3**

**Rating:** 6

**Review:**

This paper evaluates the empirical power of neural tangent kernel (NTK) on small-data tasks. The authors demonstrate the superior performance of NTK for classification/regression tasks on UCI database, small CIFAR-10 dataset and VOC07 testbed.

Overall, this paper is well written and organized. The experimental results are also quite interesting. Besides, some questions and comments are as follows:

One of the baseline algorithms in Table 1 is NN with NTK initialization. However, this paper does not give the formal definition of NTK initialization.

In Figures 1-2, it can be observed that NTK cannot universally outperform baselines on all dataset. For some dataset, NTK can be worse than baselines but for some other dataset, NTK can be significantly better than baselines. Therefore, I would like the authors to briefly discuss which kind of data can be more efficiently learned through NTK or other training algorithms.

In Tables 2-5, it can be observed that for CIFAR10 dataset, increasing the number of layers leads to higher test accuracy. But for VOC07, one can observe the opposite thing. Is there any explanation for this phenomenon?

The authors should provide a clear description of the experimental setting. For example, do you use batch normalization/weight decay in ResNets? For training NN, which optimization algorithms do you use? Do you use learning rate decay?

======================
After reading authors' response:

Thanks for your response, I would like to keep my score.


**Experience Assessment:**

I have published one or two papers in this area.

**Review Assessment: Checking Correctness Of Derivations And Theory:**

I carefully checked the derivations and theory.

**Review Assessment: Checking Correctness Of Experiments:**

I carefully checked the experiments.

**Review Assessment: Thoroughness In Paper Reading:**

I read the paper at least twice and used my best judgement in assessing the paper.

---

> ### Author Response · Authors · 2019-11-15
> **Response**
>
> Thank you for your positive review. Please find our response to your comments.
> 1.	NTK initialization means a neural network with parameterization defined in Equation 2 with all weighted being initialized to be i.i.d. $\mathcal{N}(0, 1)$. We have added a sentence after Equation 2 to clarify this.
> 2.	“In Figures 1-2, it can be observed …..” There is no clear trend on which dataset NTK can be better than other classifiers. We believe that investigating on which dataset NTK gives better performance requires more domain knowledge. Some analyses on pairwise comparisons: NTK vs. RF and NTK vs. Gaussian kernel, are provided in Section 4.2.
> 3.	“In Tables 2-5, it can be observed that ……” Note for Tables 2-5, CNTKs are used on top of raw images, so to achieve better performance, one needs to use multi-layer CNTKs to extract higher-level features. On the other hand, CNTKs on VOC07 are used on top of extracted features from ResNet-50, which are already high-level features. Therefore, shallow CNTKs suffice for this case.
> 4.	We have stated the experiment details in the third paragraph in Section B.

---

### Official Review · AnonReviewer2 · 2019-10-23
**Official Blind Review #2**

**Rating:** 8

**Review:**

This paper conducts very interesting and meaningful study of kernels induced by infinitely wide neural networks on small data tasks. They show that on a variety of tasks performance of these kernels are superior to both finite neural networks and Random Forest methods.

While neural tangent kernel (NTK) [1] is motivated for studying training dynamics of neural networks, it is also important to ask to find utility of these new powerful kernels that captures functional priors of neural networks. This paper conducted important study on small dataset regime and on a wide range of tasks (90 UCI datasets, small subset of CIFAR-10, few shot image classification task on VOC07.

Authors introduce a family of generalized NTK kernels interpolating between NNGP kernels [2] to original NTK[1] by fixing first L’ layers and allowing to train remaining layers. Treating L’ as a hyperparameter, the authors try both NNGP/NTK and kernels in between as well.

Another contribution I observe is applying kernel SVM where one utilizes NTK and shows that it can work well. This paper shows that kernels induced by infinitely wide networks could become useful for real world applications where data size is not so large.

There are few small concerns regarding experiments which are discussed in detailed comments. Overall I think the message of the paper is clear and well supported therefore I recommend accepting the paper.

Detailed comments

1) From reading the paper it was not easy to grasp where point 4 of the abstract was based on.
2) In the first footnote, small nit is that, in practice one should not invert matrix but just do a linear solve for better numerical stability and efficiency (still O(N^3) but with better constant)
3) In section 3, there seems to be no bias. Are NTK and NNs considered in this work contain no bias? Or is bias ignored for ease of presentation?
4) Nit p4 first paragraph in section 4 : multiplayer -> multilayer
5) Regards to NTK initialization performing better than standard He initialization: It was observed in [3] that for multilayer perceptron both parameterization is on-par but for CNN or WideResNet case standard parameterization performed significantly better.
6) Note that similar to analysis in section 5,  for CIFAR-10 with fully connected model [1] shows that for all dataset size(100-45k) NNGP performs better than trained neural networks.
7) One may worry that ResNet-34 is not properly tuned as most hyperparameters were fixed for large dataset.
8) Regards to hyperparameters for NTK, is there a consistent trend one could find regards to L’? What percentage of tasks that NTK performed well actually have a high L’?
9) To help the readers, I would suggest adding a little more description on statistics used for comparison as well as what VOC07 task entails.

[1] Jacot et al., Neural Tangent Kernel: Convergence and Generalization in Neural Networks, NeurIPS 2018
[2] Lee et al., Deep Neural Networks as Gaussian Processes, ICLR 2018
[3] Park et al., The Effect of Network Width on Stochastic Gradient Descent and Generalization: an Empirical Study, ICML 2019

EDIT AFTER AUTHOR RESPONSE:
I have read the response from authors. I appreciate all the efforts to improve the paper.


**Experience Assessment:**

I have published in this field for several years.

**Review Assessment: Checking Correctness Of Derivations And Theory:**

N/A

**Review Assessment: Checking Correctness Of Experiments:**

I assessed the sensibility of the experiments.

**Review Assessment: Thoroughness In Paper Reading:**

I read the paper thoroughly.

---

> ### Author Response · Authors · 2019-11-15
> **Response**
>
> Thank you for your positive review. We have revised our paper according to your comments. Please find our response to your questions below.
> -	“point 4 in the abstract”:
> For point 4, we mainly refer to Figure 2(b). We found "There is no dataset the one classifier is significantly better than the other".
> -	“Bias in NTKs and NNs”:
> We did not add bias in NTKs and NNs.
> -	“ResNet-34 is not properly tuned”
> We agree but note that there is not good way to tweak large nets on small datasets. Also note in the small data regime ($n=10$ to $n=320$), CNTK with 5,8,11 and 14 layers all beat ResNet.
> -	“is there a consistent trend one could find regards to $L’$”?
> We did not find a consistent trend. We did not try very deep NTKs ($L \le 5$ and $L’ \le 4$ in our experiments).

---

### Author Response · Authors · 2019-11-15
**General Response and Revision Summary**

We thank all reviewers for the positive reviews.
We have revised our paper to fix typos and add clarifications according to reviewers’ suggestions.

---

### Decision · Program_Chairs · 2019-12-19

**Decision:**

Accept (Spotlight)

**Comment:**

This paper carries out extensive experiments on Neural Tangent Kernel (NTK) --kernel methods based on infinitely wide neural nets on small-data tasks. I recommend acceptance.